# Effects of Film Thickness on the Residual Stress of Vanadium Dioxide Thin Films Grown by Magnetron Sputtering

**DOI:** 10.3390/ma16145093

**Published:** 2023-07-19

**Authors:** Yuemin Wang, Xingang Li, Xiangqiao Yan, Shuliang Dou, Yao Li, Lei Wang

**Affiliations:** 1Shenzhen Key Laboratory of Polymer Science and Technology, College of Materials Science and Engineering, Shenzhen University, Shenzhen 518060, China; 2College of Physics and Optoelectronic Engineering, Shenzhen University, Shenzhen 518060, China; 3Jiangxi Construction Engineering (Group) Construction Industry Investment Corporation Limited, Nanchang 330029, China; 4Jiangxi Construction Engineering (Group) Corporation Limited, Nanchang 330029, China; 5Center for Composite Materials and Structure, Harbin Institute of Technology, Harbin 150080, Chinayaoli@hit.edu.cn (Y.L.)

**Keywords:** residual stress, vanadium dioxide, film thickness, surface morphology, growth stress

## Abstract

Vanadium dioxide (VO_2_) thin films of different thicknesses were prepared by regulating the deposition time (2, 2.5, 3, and 3.5 h). The impact of deposition time on the microstructure, surface morphology, and cross-section morphology was investigated. The results showed that the grain size increased with the film thickness. Meanwhile, the influence of film thickness on the residual stress was evaluated by X-ray diffraction. The phenomenon of “compressive-to-tensile stress transition” was illustrated as the thickness increased. The change of dominant mechanism for residual stress was used for explaining this situation. First, the composition of residual stress indicates that growth stress play a key role. Then, the effect of “atomic shot peening” can be used to explain the compressive stress. Lastly, the increased grain size, lower grain boundary density, and “tight effect” in the progress of film growth cause tensile stress.

## 1. Introduction

Vanadium dioxide (VO_2_) undergoes an invertible semiconductor–metal transition (SMT) at approximately 68 °C, together with dramatic variations in optical and electrical properties [1]. Based on these unique phenomena, VO_2_ is widely used in thermal switching [2], smart windows [3,4], smart thermal control coating [5,6,7,8], infrared stealth devices [9], and so on. Different deposition methods may affect the morphology, structure, and other characteristics. According to the reaction medium involved, deposition methods can be divided into gas- and solution-based deposition [10]. Gas-based deposition is mainly composed of chemical vapor deposition (CVD) [11] and physical vapor deposition (PVD) [12]. Solution-based deposition includes sol–gel [13], the electrochemical process [14], the hydrothermal process [15], and polymer-assisted deposition [16]. The sol–gel process is a typical wet chemical technique, and due to its fine control ability of chemical compositions, it has been extensively used to synthesize films doped with other elements. Polymer-assisted deposition is a versatile strategy for the deposition of high-quality polycrystalline films, providing a cheap and scalable alternative method for the sol–gel process. The hydrothermal method is an approach to fabricating nano-scale thin films and has the advantage of precise phase control. The hydrothermal product is a metastable or mixture phase, which makes the preparation of specific thermochromic VO_2_ films more difficult. The electrochemical approach has the ability to deposit thin films on substrates with complex geometries. It requires a conducting substrate, which is very helpful for device fabrication. CVD is a common industrial technique for depositing high-performance thin films, but it has the limitations of vapor transport and the equipment required is usually complex and expensive. Compared to other methods, the competitive advantages of the magnetron sputtering process include remarkable homogeneity and compactness in products, promising scalability in large-scale substrates, and high efficiency in deposition.

In practical terms, thin films are mostly used in the formation of films/substrates. Superimposed differences in preparation conditions and methods result in thin film/substrate materials in a complex stress state, leading to inevitable residual stress during the preparation process [17,18,19]. Residual stress is the main reason for functional failure. Generally, tensile stress triggers cracking, while compressive stress triggers wrinkles and bubbling [20]. These failures destruct the configurational integrity and physical properties of thin films. At present, research on residual stress for thin films is facing many difficulties. On the one hand, it is hard to interpret the generation mechanism. The stress originates from some structural imperfections (such as impurities, vacancies, grain boundaries, dislocations, and stacking faults), surface energy, and lattice mismatch [21,22]. It involves complex physical and chemical processes and relates to nucleation and grain growth. Different growth stages or deposition conditions can lead to different stress mechanisms playing a leading role. The exploration of the mechanism of residual stress generation is still in an early stage. Existing mechanism models include the impurity effect [23], lattice mismatch [24], atomic shot peening [25], and grain boundary elimination [26]. On the other hand, there are many factors during film preparation and post-treatment that affect the stress. For example, the influence of characteristic parameters (reaction conditions [27], substrates [28], annealing temperature [29], and thickness [30]) has become a research focus. Asa’ad et al. evaluated the influence of sputtering power, sputtering pressure, and annealing time on the residual stress of sputtered tantalum thin films [31]. Kusaka et al. investigated the crystal orientation and residual stress development in AlN films at various input powers [32]. Liu et al. studied the impact of substrate temperature on the structural and attribute evolution of magnetron-sputtered Ti6Al4V films [33]. Unfortunately, there are few reports on the systematic study of the residual stress of VO_2_ thin films, and only some qualitative speculation exists, such as the effect of phase transition [34,35,36,37]. In addition, the test method is also a key point in the study of the residual stress of thin films. Traditional measurement methods such as drilling, ring strain relief, and ultrasound cannot meet the testing requirements of micro-nano-scale thin films [38]. For instance, the installation and alignment of the sample is very difficult, and the resolution cannot be satisfied. Moreover, size and surface effects make analysis more complex. Therefore, some calculation methods are continuing to be improved. For example, the curvature method based on Stoney’s formula is suitable for measuring almost all types of thin films [39], while the X-ray diffraction technique can be used on typically crystalline thin films [40], and the Raman spectroscopy is used for amorphous thin films [41].

In this paper, VO_2_ thin films of different thicknesses were prepared by magnetron sputtering. The impact of film thickness, grain size, microstructure, and morphology on residual stress was explored. At the same time, the dominant mechanism of residual stress is discussed in detail.

## 2. Experiment Details

VO_2_ thin films were deposited on ITO substrates by magnetron sputtering with a metal vanadium target (99.99%, Φ 76.2 mm). A turbo-molecular pump was used to realize a base vacuum of less than 5 × 10^−3^ Pa. A mixture of Ar (99.9999% purity) and O_2_ (99.9999% purity) was introduced into the chamber. The main fabrication parameters are shown in Table 1. The deposition time was changed to regulate the thickness of the films. Then, the samples were heat-treated at 400 °C for 2 h in order to obtain high crystalline VO_2_.

The crystalline structures of the samples were characterized using an X-ray diffraction diffractometer (Empyrean, Malvern Panalytical, Almelo, Netherlands) with Al-K*α*radiation. The vibrational modes were determined using Raman microscopy (XploRA PLUS, HORIBA Scientific, Montpellier, France) with a 532 nm laser. The film morphology was observed through an atomic force microscope (Dimension3100, BRUKER, Bremen, Germany), along with scanning electron microscopy (Supra, Zeiss, Oberkochen, Germany). The reflectance of the samples was characterized using infrared spectroscopy (VERTEX-70, BRUKER, Bremen, Germany).

In this paper, the X-ray diffraction approach was employed to test residual stress [42]. X-ray diffraction is an important, attractive, and nondestructive measurement technique that allows evaluating the average residual stress. This method consists of measuring the lattice spacing of a specific plane at different tilt angles. According to the theory of elasticity and Bragg diffraction, the residual stress can be obtained by Equation (1):(1)σ=K⋅M
where $M=\partial (2\theta )/\partial (\sin^{2}\psi )$, $K=-E/2(1+\upsilon )\cot\theta_{0}$, K is the stress constant, $\theta_{0}$ is the diffraction half-angle without stress, and ψ is the angle between the normal from the sample surface and the normal from the diffractive crystal surface.

## 3. Results and Discussion

### 3.1. Structural Characterization

Figure 1a indicates the Raman spectroscopy for the VO_2_ thin films. In the case of different deposition times, obvious Raman bands were discovered at 190, 221, 304, 385, 504, and 610 cm^−1^. In general, the intensity and width of the Raman peaks were weak and wide for thinner films. However, the phenomenon did not exist in this work, because the four kinds of films were relatively thick. The peak locations, symmetry assignments, and spectral characteristics are similar to those in other studies, and proved high crystallinity of thin films [43,44,45]. It is worth noting that there was no ITO Raman peak in this spectroscopy, because VO_2_ has high absorption for 532 nm wavelength.

Figure 1b shows the infrared reflectance spectra that formed for VO_2_ thin films under semi-conductive and metallic states. In the cold state (insulating, M-phase, a monoclinic-like structure), the VO_2_ thin films indicated high infrared transmittance. After interference with ITO, wave troughs formed. With the increase in deposition time, the wave trough appeared red-shifted and the modulation amplitude started to increase. After the phase transition (metallic, R-phase, a rutile-like structure), the VO_2_ thin film changed from high infrared transmission to high reflection. These changes also indicate that high-performance VO_2_ thin films had been prepared.

Figure 2a shows the XRD patterns for VO_2_ thin films with different deposition times. Diffraction peaks located at 27.81°, 55.56°, and 70.58° can clearly be observed. These conditions were regarded as the VO_2_ (011), (−213), and (−132) planes (JCPDS 43-1051). The (011) peaks were far more powerful than others, corresponding to other reports [43,46]. The intensity of the VO_2_ (011) peaks showed an acute growing tendency as the deposition time increased. This demonstrates that the crystallinity of VO_2_ thin films can be improved through increasing the thickness. In addition, no other impurity peaks were observed except for ITO, indicating the formation of high-performance thin films.

Figure 2b shows the grain sizes and full width at half-maximum (FWHM) of VO_2_ (011) peaks with different deposition times. It can be concluded that the grain size increased with increasing deposition time. As the thickness increased, the full width at half-maximum decreased from 0.299° to 0.197°, while the grain sizes increased from 27.11 to 41.14 nm, calculated using the Scherrer formula. Similar phenomena have also been reported [43,47,48].

### 3.2. Morphology

Figure 3 indicates the surface morphology of the VO_2_ thin films with various deposition times. It can be seen that the surface of all films were predominantly flat without obvious cracks, revealing that the surface morphology was independent of thickness in this research.

Figure 4 shows the cross-section morphology of the VO_2_ thin films on ITO substrates with various deposition times. The cross-section images were used to determine the film thickness and analyze the process of growth. The films were homogeneous and dense and for all films, the morphology can be characterized as compact packing of columns. Similar columnar-like growth can also be observed for the VO_2_ films deposited by magnetron sputtering in other research works [43,47]. The film thickness was found to vary from 460 to 720 nm as the deposition time increased from 2 to 3.5 h.

Figure 5 shows AFM images of the surface morphology of the VO_2_ films with four different thicknesses. Under different conditions, all of the films exhibited a flat surface and tightly packed morphology, which is similar to the phenomenon observed by SEM. The average surface roughness, measured by AFM, was approximately 4.3 nm (more details in the Appendix A) (more details in the Appendix A).

### 3.3. Residual Stress

The X-ray diffraction method was performed to measure the residual stress (more details in the Appendix A). The residual stress obtained is plotted in Figure 6 as a function of deposition time. The thinnest film had the highest compressive stress of −9.009 GPa, while the film resulting from a 3.5 h deposition time had the highest tensile stress of 18.42 GPa. This indicates that as the film thickness increased, the residual stress presented the phenomenon of “compression-to-tensile stress transition.”

According to the deposition and formation process of thin films, the residual stress consists of thermal stress, interface stress, and growth stress [49,50,51]. Growth stress comes from structural drawbacks in the growth progress, while thermal and interface stresses occur due to mismatch of the thin film and substrate. First, at the initial stage of film deposition, if there is a large difference between the film and substrate or quite a high density of defects and impurities, interface mismatch occurs and leads to interface stress. However, previous work has shown that when the film thickness in the range of sub-microns, interface stress does not play a critical role in the total residual stress [46].

Second, thermal stress is generated when annealing due to mismatch of the CTE (coefficient of thermal expansion) between the film and the substrate. On the basis of the analytical model proposed by Tsui (Appendix A) [50], thermal stress is the function of temperature, thickness, elastic modulus, and CTE. According to the model, when the thickness of the thin film is far less than the substrate, the film thickness has little effect on the thermal stress. Thermal stress in this study was simulated, and it can be seen in the Appendix A. The results revealed that the thermal stress ranged from −0.21 to 0.023 GPa, so the impact on the residual stress can be ignored. Therefore, the change of residual stress in this work should be considered the influence of growth stress.

The formation of growth stress involves complex physical and chemical processes, which are the result of multiple mechanisms. As shown in Figure 6, residual stress can be divided into two parts: Compressive stress and tensile stress. In the first part, the “atomic shot peening” effect is the main control mechanism [51], which is linked to the clash from neutral inert gas atoms as the film grows. In the progress of sputtering, these adatoms from the film surface are hit by the incident ions and become imbedded in film’s sub-surface through knock-on. The misfitted atoms produce a strain area in the circumambient matrix, thus bringing about a kind of compressive stress among films.

When the film growth proceeds to a certain stage, the dominating mechanism is changed. The transition from compressive to tensile stress may be interpreted as being caused by two reasons. On the one hand, in the process of film deposition, the surface atoms do not have enough time to diffuse, and the metastable structure undergoes a spontaneous ordering process. Ordered atoms, along with the annihilation of pores and drawbacks, bring about volume shrinkage and densification of thin films, triggering a natural trend of the films for the sake of promoting tensile stress owning the film thickness. When the tensile stress of this “tight effect” exceeds the compressive stress caused by the “atomic shot peening” effect, tensile stress gradually occurs. On the other hand, the grain size increases with the film thickness, and the density of grain boundaries decreases. In this case, there is less probability of excess atoms occurring for the purpose of being inserted into the grain boundaries, so the compressive stress decreases and gradually transforms into tensile stress. To sum up, the thickness not only affects the grain size, but also causes the transition of the dominant mechanism for residual stress.

## 4. Conclusions

In conclusion, VO_2_ thin films of different thicknesses deposited onto ITO substrates were prepared by direct current magnetron sputtering. The results proved that the grain size increases with the film thickness. The values of residual stresses varied from the compressive to tensile state as the thickness increased. On the one hand, the “atomic shot peening” effect can be used to explain the compressive stress. On the other hand, the increased grain size, lower grain boundary density, and “tight effect” by film growth can cause the tensile stress. The transition of the dominant mechanism for residual stress was used to explain the phenomenon of “compressive-to-tensile stress transition.” The results of this study could be beneficial for the design and application of many functional materials.

## Figures and Tables

**Figure 1 materials-16-05093-f001:**
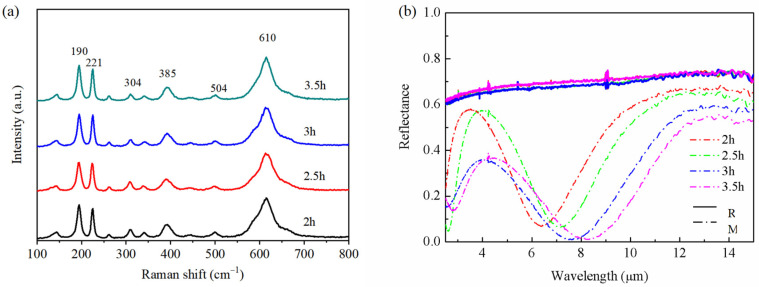
(**a**) Raman spectra of the VO_2_ thin films with different deposition times. (**b**) The reflectance spectra of VO_2_ thin films with different deposition times.

**Figure 2 materials-16-05093-f002:**
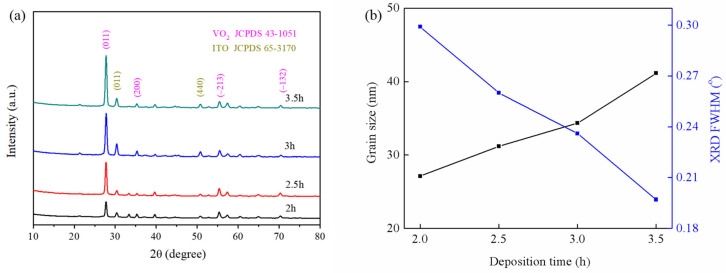
(**a**) XRD patterns of the VO_2_ thin films with different deposition times. (**b**) Grain sizes and XRD FWHM of the VO_2_ peaks with different deposition times.

**Figure 3 materials-16-05093-f003:**
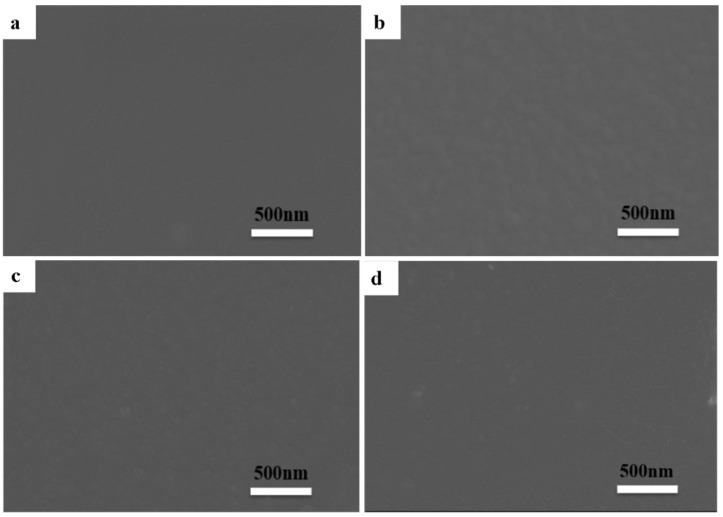
The surface morphology of the VO_2_ thin films with various deposition times: (**a**) 2 h, (**b**) 2.5 h, (**c**) 3 h, and (**d**) 3.5 h.

**Figure 4 materials-16-05093-f004:**
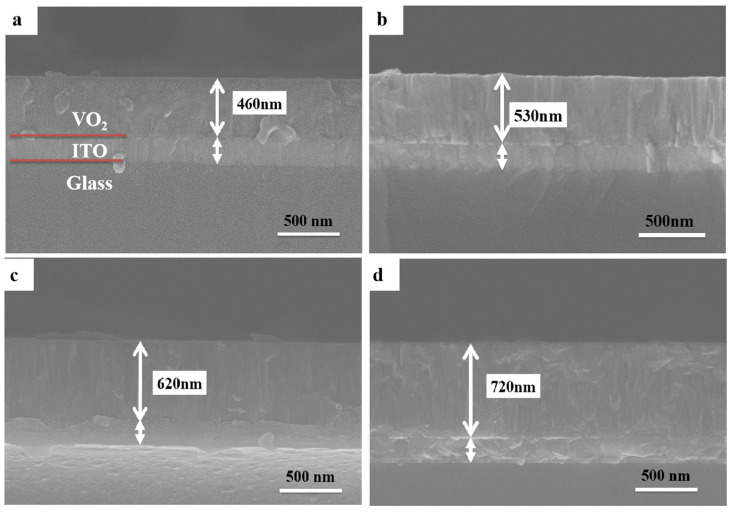
The cross-section morphology of the VO_2_ thin films on ITO substrates with various deposition times: (**a**) 2 h, (**b**) 2.5 h, (**c**) 3 h, and (**d**) 3.5 h.

**Figure 5 materials-16-05093-f005:**
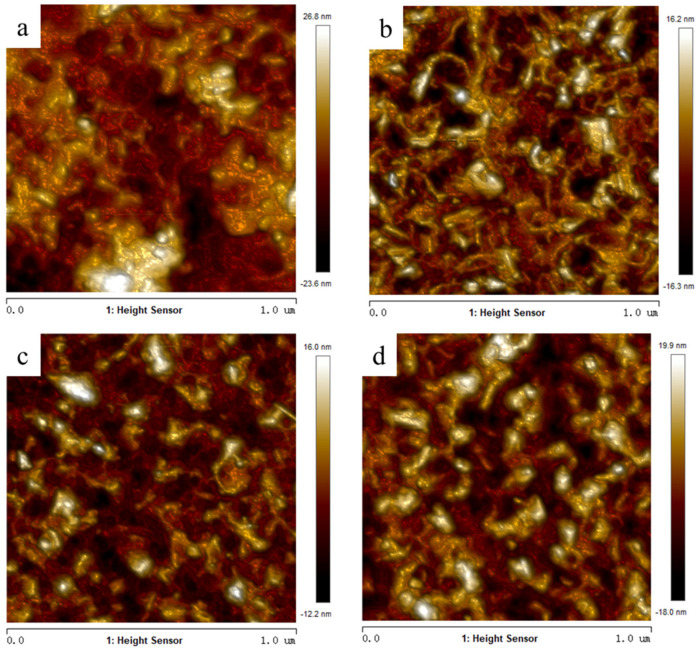
AFM images of the VO_2_ thin films with various deposition times: (**a**) 2 h, (**b**) 2.5 h, (**c**) 3 h, and (**d**) 3.5 h.

**Figure 6 materials-16-05093-f006:**
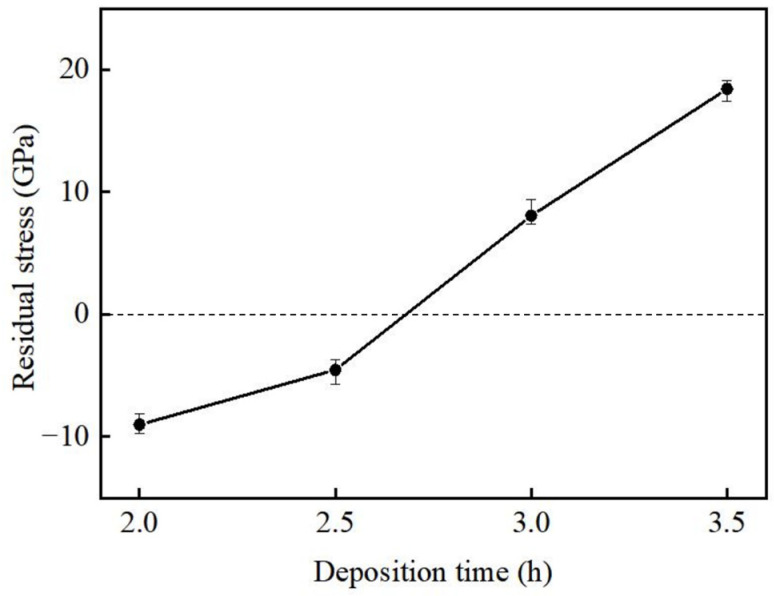
The residual stress of VO_2_ thin films with different deposition times.

**Table 1 materials-16-05093-t001:** The main fabrication parameters in the experiment.

Samples	Temperature (°C)	Time (h)	Pulse Width (μs)	Pulse Frequency (Hz)	Pressure (Pa)	Ar:O_2_ Ratio (sccm)	Average Power (W)
S1	400	2.0	50	200	0.9	80:1.4	200
S2	400	2.5	50	200	0.9	80:1.4	200
S3	400	3.0	50	200	0.9	80:1.4	200
S4	400	3.5	50	200	0.9	80:1.4	200

## Data Availability

All data presented in this study are available upon request from the corresponding author.

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
