# Peer review of "Effects of Film Thickness on the Residual Stress of Vanadium Dioxide Thin Films Grown by Magnetron Sputtering"

_materials, 2023, doi:10.3390/ma16145093_

Round 1

Reviewer 1 Report

The manuscript “Effects of film thickness on the residual stress of vanadium dioxide thin films grown by magnetron sputtering” describes the deposition process and characterization of VO2 thin films.

There are some major points to be addressed before considering publication of this manuscript, as follows:

-there are only 4 samples deposited, at same conditions, only different deposition times. The choice of these particular thicknesses is not clear. It is generally known that by increasing the thickness the stress increases, and there is a limit of thickness that can be supported by a layer.

- it is not clear why this annealing temperature and time were chosen and whether had been optimized for obtaining this degree of crystallinity or not.

-in the introduction it is stated that “there are no reports for systematic study on the residual stress of VO2 thin film, and only some qualitative speculation exists”. One quick survey of literature reveals that there are studies of this type, such as: Journal of Materials Research , Volume 26 , Issue 11 , 14 June 2011 , pp. 1384 – 1387 DOI: https://doi.org/10.1557/jmr.2011.134;  Journal of Materials Research Volume 19 Issue 8 , August 2004 , pp. 2306 – 2314 DOI: https://doi.org/10.1557/JMR.2004.0299, https://doi.org/10.1016/j.scriptamat.2010.11.018, etc

-in the experimental details some of the key parameters are missing, such as: gas flow ratio O2/Ar, distance from vanadium target to substrate holder; this section could be further improved by including more details about the used characterization techniques.

-most of the figures are of poor quality and the text is hard to read, even by zooming.

-most of the results from Raman, XRD, SEM and AFM measurements showed no significant differences between samples or clear trends.

-in section 3.2 there is a whole discussion about “particles” that are accumulating. What kind of particles are those? It is not a continuous film? This assumption with particles is not supported by corresponding cross section images or grain size evaluation from XRD measurements; similarly, the Authors mentioned “competitive growth” without elaborating on the concept or how it relates to the results.

- the figure 2 legend is misleading, because the metal-insulator transition (MIT) was not properly investigated and discussed; also, the effect of annealing temperature on the samples structure is evidenced only by the spectrophotometric measurements; moreover, the choice of using logarithmic scale on Y axis is unusual.

- the thickness of ITO layer deposited on glass substrate is not provided and it is hard to be estimated from the inset within Figure 3; also, the optical, structural and morphological properties of ITO layers are important and should be separately discussed.

-in the discussions of AFM images there is again a reference to grain size. There are no visible grains on the surface, just a continuous layer with a certain roughness. There is no quantitative evaluation of these grains.

- the evaluation of stress should be more clearly detailed. The reference cited [41] refers to a multilayer structure.

-extensive English editing should be performed. In the manuscript there are some unusual uses of words and expressions. A few examples in the following:

“thin films owning the thicknesses” in the abstract

“the film thickness ascends.” In the abstract and in the text.

Introduction:

“Solution-deposition is mainly consists”

“on residual stress is the research highlights.”

“thin films owning disparate thicknesses”

2. Experimental details:

“The corresponsive sputtering”

Results and discussions:

‘peaks are far powerful compared to others,’

“crystallized VO2 thin film has been wonderfully fabricated.” Sounds a bit exaggerated

3.3. Residual stress

“Having learning the structure”

“ascended to an affirmatory value”

The English used in some parts of the manuscript is unclear and subjective (“the greatly-cristallized VO2 thin films has been wonderfully fabricated”) and does not provide specific and precise scientific information.

Overall, the manuscript lacks sufficient scientific rigor and clarity, and therefore, I would recommend rejecting it.

Extensive English editing should be performed. In the manuscript there are some unusual uses of words and expressions. Some examples are given in the report

Author Response

The manuscript “Effects of film thickness on the residual stress of vanadium dioxide thin films grown by magnetron sputtering” describes the deposition process and characterization of VO2 thin films.There are some major points to be addressed before considering publication of this manuscript, as follows:

  • there are only 4 samples deposited, at same conditions, only different deposition times. The choice of these particular thicknesses is not clear. It is generally known that by increasing the thickness the stress increases, and there is a limit of thickness that can be supported by a layer.

Response: Thanks for the reviewer’s comment. Firstly, the reasons for selecting these particular thicknesses in this article are: 1)Young’s modulus is the key property for calculating residual stress, which affects measurement accuracy. The Young’s modulus present obvious size effect at the micro-nano scale, so the thickness selected in this study are all sub-micron scale in order to alleviate the influence of size effect. 2)To ensure testing accuracy and obtain high-quality diffraction intensity and peaks, the author selected a thicker film compared to other studies. Secondly, the purpose of this paper mainly studies the relationship between thickness, grain size, micro-structure and residual stress. The results of four samples have proved the relationship and found the phenomenon of the “compressive-to-tensile stress transition”.

  • it is not clear why this annealing temperature and time were chosen and whether had been optimized for obtaining this degree of crystallinity or not.

Response: Thanks for the reviewer’s comment. It is well known that the fabrication parameters of VO2 films are rigorous since the plenty of Vanadoxy compound. In order to study the effect of film thickness on the residual stress of VO2 thin films, we selected mature process parameters (as shown in Table 1) and characterized their crystallinity through XRD.

  • in the introduction it is stated that “there are no reports for systematic study on the residual stress of VO2 thin film, and only some qualitative speculation exists”. One quick survey of literature reveals that there are studies of this type, such as: Journal of Materials Research , Volume 26 , Issue 11 , 14 June 2011 , pp. 1384 -1387 DOI: https://doi.org/10.1557/jmr.2011.134; Journal of Materials Research , Volume 19,Issue8,August2004,pp.2306-2314DOI:https://doi.org/10.1557/JMR.2004.0299,https://doi.org/10.1016/j.scriptamat.2010.11.018, etc?

Response: Thanks for the reviewer’s comment. We had changed the description in the introduction and the references were added.

“there are a few reports for systematic study on the residual stress of VO2 thin film, and only some qualitative speculation exists, such as the affect of phase transition [34-37].”

References:

[35]Balakrishnan V, Ko C, Ramanathan S. Size effects on stress relaxation across the metal-insulator transition in VO2 thin films[J]. Journal of Materials Research, 2011, 26(11): 1384-1387.

[36]Tsai K Y, Chin T S, Shieh H P D, et al. Effect of as-deposited residual stress on transition temperatures of VO2 thin films[J]. Journal of materials research, 2004, 19(8): 2306-2314.

[37]Viswanath B, Ko C, Ramanathan S. Thermoelastic switching with controlled actuation in VO2 thin films[J]. Scripta Materialia, 2011, 64(6): 490-493.

  • in the experimental details some of the key parameters are missing, such as: gas flow ratio O2/Ar, distance from vanadium target to substrate holder; this section could be further improved by including more details about the used characterization techniques.  

Response: Thanks for the reviewer’s comment. We had added the experimental details.

“VO2 thin films were deposited on ITO substrates by magnetron sputtering with a metal vanadium target (99.99%, Φ76.2 mm). The turbo-molecular pump was used to realize a base vacuum of less than 5 × 10−3 Pa. And the mixture of Ar (99.9999% purity) and O2 (99.9999% purity) was introduced into the chamber. The main fabrication parameters are shown in Table 1. The deposition time was changed to regulate the thickness of films. And then, the samples would be heat treated at 400℃ for 2h in order to obtain the high crystalline VO2. ”

Table 1 The main fabrication parameters in the experiment

Samples

Temperature

(℃)

Time

(h)

Pulse width(μs)

Pulse frequency

(Hz)

Pressure

(Pa)

Ar:O2 Ratio (sccm)

Average

power

(W)

S1

400

2.0

50

200

0.9

80:1.4

200

S2

400

2.5

50

200

0.9

80:1.4

200

S3

400

3.0

50

200

0.9

80:1.4

200

S4

400

3.5

50

200

0.9

80:1.4

200

  • most of the figures are of poor quality and the text is hard to read, even by zooming.

Response: Thanks for the reviewer’s comment. The quality of figures had been changed to add the readability.

  • most of the results from Raman, XRD, SEM and AFM measurements showed no significant differences between samples or clear trends.

Response: Thanks for the reviewer’s comment. The corresponding results had been re-discussed in the revision.

  • in section 3.2 there is a whole discussion about “particles” that are accumulating. What kind of particles are those? It is not a continuous film? This assumption with particles is not supported by corresponding cross section images or grain size evaluation from XRD measurements; similarly, the Authors mentioned “competitive growth” without elaborating on the concept or how it relates to the results.

Response: Thanks for the reviewer’s comment. The discussion had been modified. And the description of “particles” had been replaced by the surface roughness. It is undoubtedly the samples are continuous film.  

  • the figure 2 legend is misleading, because the metal-insulator transition (MIT) was not properly investigated and discussed; also, the effect of annealing temperature on the samples structure is evidenced only by the spectrophotometric measurements; moreover, the choice of using logarithmic scale on Y axis is unusual.

Response: Thanks for the reviewer’s comment. The figure had been changed. It can be seen obvious metal-insulator transition.

  • the thickness of ITO layer deposited on glass substrate is not provided and it is hard to be estimated from the inset within Figure 3; also, the optical, structural and morphological properties of ITO layers are important and should be separately discussed.

Response: Thanks for the reviewer’s comment. The ITO glass had been used as substrate. The thickness of ITO layer can be seen in cross-section of samples(as shown in Figure 4). And the discussion for ITO layer had been added.

“In addition, it can be seen that no other impurity peak was observed for the four samples except for ITO peaks, indicating the formation of high-performance thin films.”

  • in the discussions of AFM images there is again a reference to grain size. There are no visible grains on the surface, just a continuous layer with a certain roughness. There is no quantitative evaluation of these grains.

Response: Thanks for the reviewer’s comment. The grain size had been replaced by surface roughness.

  • the evaluation of stress should be more clearly detailed. The reference cited [41] refers to a multilayer structure.

Response: Thanks for the reviewer’s comment. This literature was cited to illustrate the principle of X-ray diffraction technique. A new literature was added to avoid controversy, and the previous one has deleted.

[42]Ali A, Chiang Y W, Santos R M. X-ray diffraction techniques for mineral characterization: A review for engineers of the fundamentals, applications, and research directions[J]. Minerals, 2022, 12(2): 205.

  • extensive English editing should be performed. In the manuscript there are some unusual uses of words and expressions. A few examples in the following:

“thin films owning the thicknesses” in the abstract;“the film thickness ascends.” In the abstract and in the text. Introduction:“Solution-deposition is mainly consists”;“on residual stress is the research highlights.”;“thin films owning disparate thicknesses” 2. Experimental details:“The corresponsive sputtering”; Results and discussions:‘peaks are far powerful compared to others,’ “crystallized VO2 thin film has been wonderfully fabricated.” Sounds a bit exaggerated; 3.3. Residual stress;“Having learning the structure”;“ascended to an affirmatory value”;

Response: Thanks for the reviewer’s comment. The further english editing had been performed.

  • The English used in some parts of the manuscript is unclear and subjective (“the greatly-cristallized VO2 thin films has been wonderfully fabricated”) and does not provide specific and precise scientific information.

Response: Thanks for the reviewer’s comment. The English editing had been performed. And the specific and precise scientific information of manuscript had been strengthened.

Reviewer 2 Report

The issue addressed in the article is exciting and undoubtedly provides research value. However, it is regrettable that the authors did not present it in a slightly broader context of mechanical properties investigations. Overall, the most significant merit of the text is the reliability and thorough attempt to interpret the residual results. Its weaknesses are the sparsity of the research methods, imprecise wording, and questionable theses based on limited research material. However, I think that the text as it is can be improved. Below, the authors will find a list of my comments.

1.      In the opening section of the Introduction, the authors list the methods of fabricating VO2 films in the literature. Can you briefly characterize their effectiveness? What are their main advantages and disadvantages? How does the magnetron sputtering method compare in this regard?

2.      Experiment details

Was there a specific reason why ITO substrates were used?

3.      Experiment details – "More preparation details can be refer to our previous research [39-40]"

Ref. 40 does not refer to VO2 fabrication by magnetron sputtering. Ref. 39 describes an experiment in which the HiPIMS system was used. In the Conclusions section, meanwhile, the authors mention that they used the dc-MS technique. This is very misleading and should be clarified. In my opinion, the characteristics of the deposition process should be mentioned in the text. Referring to other works is very bothersome for the reader.

4.      Experiment details – "Meanwhile, the magnetron power was set at 200 W (…)"

If the authors used HiPIMS, then why is the power so low? Is it the effect of integrating the power pulse over a wide range of time? With HiPIMS, it is better to report the voltage and current parameters of the pulse. The integrated power value over a wide range is not very useful information.

5.      Experiment details – "At the stage of post-treatment, the samples would be heat treated at 400℃ for two h in order to obtain the high crystalline VO2"

Was the annealing process carried out in a vacuum chamber? In a protective gas atmosphere or air?

6.      Experiment details

XRD and Raman's spectroscopy are methods for mainly determining the phase and chemical state (Raman) of materials, not the morphology of the structure. SEM means scanning electron microscopy and not scanning electronic microscopy. Please specify the type of laser used in Raman spectroscopy, also the parameters for realizing the measurement. What was the X-ray line in the XRD method?

7.      Results and discussion

All figures in the article are of inferior quality. So low that some cannot be read. Is this due to the low resolution of these figures, or is it the result of the editorial office folding the PDF?

8.      Results and discussion – "In case of different thicknesses, Raman bands could be discovered at 190, 221, 304, 385, 504 and 610 cm-1 , respectively. This phenomenon proved high crystallinity of thin films."

On what basis do the authors believe that this form of Raman spectra indicates high crystallinity?

9.      Results and Discussion – Fig.1

Do the parameters of the peaks change depending on the thickness: position, and FWHM? If the morphology of the structure, degree of crystallinity, and stresses change, perhaps the spectra reflect this. Have any differences been observed in XRD?

10.  Results and discussion

The terms M-phase and R-phase should be explained in the text

11.  Results and discussion

Could the authors calculate the size of the crystallites from the XRD results? I think this could be useful if the authors often discuss grain sizes.

12.  Results and discussion – Fig.3

I think exposing SEM images from the surface of the samples in Fig is pointless, mainly because the authors use AFM images afterward. I would much rather see all the photos of cross-sections of the layers because there, you can see their structure. I would use the SEM photos of the surface as an insert in the figure

13.  Results and discussion – "It could be seen that as thickness ascends, surface tends to get denser and the particle size tends to ascend. (…) For another, while film thickness reaches the vital point, nucleation deposited owns an exceedingly highly surface free energy, which triggers the blend of crystal grains. By and large, the particle dimension of thin films ascended constantly as the thickness ascends, which was backed up by the consequences from XRD patterns."

I have a problem interpreting the films' surface morphology from SEM images. Films are very smooth; SEM images do not provide good contrast. I cannot define the differences in the size of the forms of their build on the surface, porosity, crystallinity, and density. I believe that the AFM studies are better about surface morphology. I would use SEM images only to present and discuss the morphology of the internal structure.

I think the authors should rethink the correct use of the term "grain." Usually, the terms "grain" and "crystallite" are synonymous. Therefore, I have doubts about whether the visible forms of the surface structure of the films are, in fact, grains. Instead, I believe they are agglomerates of grains. Comparing the calculated crystallite sizes from the XRD with the sizes of the surface forms in the AFM images should provide an answer.

14.  Fig.4

When presenting AFM results, it is better to use 2D and 3D images. The generated 3D image slightly disturbs the perspective of assessing the topography in the XY plane. I would encourage the authors to use a 2D image processing program to extract the image of structure forms on the surface and make some statistics. Assessing the size, shape, and density of structure forms from such images is very doubtful. The differences are minor. Additionally, the images lack a color scale.

15.  Conclusions – "On the other hand, the increase of grain size, better crystallinity, and the denser of surface would cause the tensile state."

This sentence sums up the weakness of the article. Grain size, crystallinity, and film density can be determined using reliable methods. Unfortunately, the authors did not use any of them

The text was understandable and quite easy for me to read, however, I noticed many linguistic errors. I think the text should be reviewed by at least a native speaker.

Author Response

  • In the opening section of the Introduction, the authors list the methods of fabricating VO2films in the literature. Can you briefly characterize their effectiveness? What are their main advantages and disadvantages? How does the magnetron sputtering method compare in this regard?

Response: Thanks for the reviewer’s comment.The corresponding descriptions had been discussed in the introduction.

“The sol-gel process is a typical wet-chemical technique, due to its fine control ability of chemical compositions, it has been extensively used to synthesize films doped with other elements. Polymer-assisted deposition is a versatile  strategy for the deposition of highquality polycrystalline films, providing a cheap and scalable alternative method for the sol-gel process. The hydrothermal method is the approach to fabricate nano-scale thin film, and has the advantage of precise phase control. The hydrothermal product is a metastable phase or mixture phase which makes the preparation of the specific thermochromic VO2 film more difficult. Electrochemical approach have the ability of depositing thin films on substrates with complex geometries. It requires the conducting substrate, which is very helpful for device fabrication. CVD is a common industrial technique for depositing high-performance thin films, but it has the limitations of vapor transport and the equipment required is usually complex and expensive. Compared to other methods, the competitive advantages of the magnetron sputtering process are remarkable homogeneity and compactness in products, promising scalability in large-scale substrates, and high efficiency in deposition.”

  • Experiment details

Was there a specific reason why ITO substrates were used?

Response: Thanks for the reviewer’s comment.The main purpose of using ITO as the substrates is to alleviate the mismatch of CTE(coefficient of thermal expansion) between the film and the substrate. Compared with other materials, the CTE of ITO is about 5.9*10-6/℃, similar to VO2(5.7*10-6/℃).The thermal stress in this study has been simulated, and the results reveal that the thermal stress can be ignored.

  • Experiment details – "More preparation details can be refer to our previous research [39-40]"Ref. 40 does not refer to VO2fabrication by magnetron sputtering. Ref. 39 describes an experiment in which the HiPIMS system was used. In the Conclusions section, meanwhile, the authors mention that they used the dc-MS technique. This is very misleading and should be clarified. In my opinion, the characteristics of the deposition process should be mentioned in the text. Referring to other works is very bothersome for the reader.

Response: Thanks for the reviewer’s comment.We had re-describled the experimental details in the article. 

“VO2 thin films were deposited on ITO substrates by magnetron sputtering with a metal vanadium target (99.99%, Φ76.2 mm). The turbo-molecular pump was used to realize a base vacuum of less than 5 × 10−3 Pa. And the mixture of Ar (99.9999% purity) and O2 (99.9999% purity) was introduced into the chamber. The main fabrication parameters are shown in Table 1. The deposition time was changed to regulate the thickness of films. And then, the samples would be heat treated at 400℃ for 2h in order to obtain the high crystalline VO2. ”

Table 1 The main fabrication parameters in the experiment

Samples

Temperature

(℃)

Time

(h)

Pulse width(μs)

Pulse frequency

(Hz)

Pressure

(Pa)

Ar:O2 Ratio (sccm)

Average

power

(W)

S1

400

2.0

50

200

0.9

80:1.4

200

S2

400

2.5

50

200

0.9

80:1.4

200

S3

400

3.0

50

200

0.9

80:1.4

200

S4

400

3.5

50

200

0.9

80:1.4

200

  • Experiment details – "Meanwhile, the magnetron power was set at 200 W (…)"

If the authors used HiPIMS, then why is the power so low? Is it the effect of integrating the power pulse over a wide range of time? With HiPIMS, it is better to report the voltage and current parameters of the pulse. The integrated power value over a wide range is not very useful information.

Response: Thanks for the reviewer’s comment. Similar to the previous question, we had added the experimental details in the article. 

  • Experiment details-"At the stage of post-treatment, the samples would be heat treated at 400℃for two h in order to obtain the high crystalline VO2"Was the annealing process carried out in a vacuum chamber? In a protective gas atmosphere or air?

Response: Thanks for the reviewer’s comment. Similar to the previous question, we had added the experimental details in the article. 

  • Experiment details

XRD and Raman's spectroscopy are methods for mainly determining the phase and chemical state (Raman) of materials, not the morphology of the structure. SEM means scanning electron microscopy and not scanning electronic microscopy. Please specify the type of laser used in Raman spectroscopy, also the parameters for realizing the measurement. What was the X-ray line in the XRD method?

Response: Thanks for the reviewer’s comment. We had changed the description in the experiment details. 

“The crystalline structures of the samples were characterized by an X-ray diffraction diffractometer(Empyrean) with Al-Kradiation.The vibrational modes were determined using Raman Microscopy (XploRA PLUS) with a 532 nm laser. The film morphology was observed through the atomic force microscope (Dimension3100), along with scanning electron microscopy (Zeiss Supra). Reflectances of samples were characterized by the infrared spectroscopy (BRUKER VERTEX-70)”

  • Results and discussion

All figures in the article are of inferior quality. So low that some cannot be read. Is this due to the low resolution of these figures, or is it the result of the editorial office folding the PDF?

Response: Thanks for the reviewer’s comment. The quality of figures had been changed to add the readability.

  • Results and discussion – "In case of different thicknesses, Raman bands could be discovered at 190, 221, 304, 385, 504 and 610 cm-1 , respectively. This phenomenon proved high crystallinity of thin films."On what basis do the authors believe that this form of Raman spectra indicates high crystallinity?

Response: Thanks for the reviewer’s comment. The related references had been added to proved this discussion.

[44]Petrov G I, Yakovlev V V, Squier J. Raman microscopy analysis of phase transformation mechanisms in vanadium dioxide[J]. Applied physics letters, 2002, 81(6): 1023-1025.

[45]Pan M, Liu J, Zhong H, et al. Raman study of the phase transition in VO2 thin films[J]. Journal of Crystal Growth, 2004, 268(1-2): 178-183.

  • Results and Discussion – Fig.1

Do the parameters of the peaks change depending on the thickness: position, and FWHM? If the morphology of the structure, degree of crystallinity, and stresses change, perhaps the spectra reflect this. Have any differences been observed in XRD?

Response: Thanks for the reviewer’s comment. We had changed the figure and added the description of FWHM and grain size. 

  • Results and discussion

The terms M-phase and R-phase should be explained in the text.

Response: Thanks for the reviewer’s comment. We had changed the description in the text. 

“In cold state (insulating, M-phase, a monoclinic phase), After the phase transition (metallic, R-phase, a rutile phase)” 

  • Results and discussion

Could the authors calculate the size of the crystallites from the XRD results? I think this could be useful if the authors often discuss grain sizes.

Response: Thanks for the reviewer’s comment. We had calculated the grain size from the XRD results. 

  • Results and discussion – Fig.3

I think exposing SEM images from the surface of the samples in Fig is pointless, mainly because the authors use AFM images afterward. I would much rather see all the photos of cross-sections of the layers because there, you can see their structure. I would use the SEM photos of the surface as an insert in the figure

Response: Thanks for the reviewer’s comment. We had added the photos of cross-sections with different deposition time . 

  • Results and discussion – "It could be seen that as thickness ascends, surface tends to get denser and the particle size tends to ascend. (…) For another, while film thickness reaches the vital point, nucleation deposited owns an exceedingly highly surface free energy, which triggers the blend of crystal grains. By and large, the particle dimension of thin films ascended constantly as the thickness ascends, which was backed up by the consequences from XRD patterns."

I have a problem interpreting the films' surface morphology from SEM images. Films are very smooth; SEM images do not provide good contrast. I cannot define the differences in the size of the forms of their build on the surface, porosity, crystallinity, and density. I believe that the AFM studies are better about surface morphology. I would use SEM images only to present and discuss the morphology of the internal structure.

I think the authors should rethink the correct use of the term "grain." Usually, the terms "grain" and "crystallite" are synonymous. Therefore, I have doubts about whether the visible forms of the surface structure of the films are, in fact, grains. Instead, I believe they are agglomerates of grains. Comparing the calculated crystallite sizes from the XRD with the sizes of the surface forms in the AFM images should provide an answer.

Response: Thanks for the reviewer’s comment. The description of “particles” had been replaced by the surface roughness. The discussion had been modified.

  • 4

When presenting AFM results, it is better to use 2D and 3D images. The generated 3D image slightly disturbs the perspective of assessing the topography in the XY plane. I would encourage the authors to use a 2D image processing program to extract the image of structure forms on the surface and make some statistics. Assessing the size, shape, and density of structure forms from such images is very doubtful. The differences are minor. Additionally, the images lack a color scale.

Response: Thanks for the reviewer’s comment. The 2D images of AFM had been replaced.

  • Conclusions – "On the other hand, the increase of grain size, better crystallinity, and the denser of surface would cause the tensile state."This sentence sums up the weakness of the article. Grain size, crystallinity, and film density can be determined using reliable methods. Unfortunately, the authors did not use any of them.

Response: Thanks for the reviewer’s comment. The discussion had been modified.

Reviewer 3 Report

It would be useful to supplement the article with data on surface roughness after AFM (Fig.4)

Figures 1 and 3 are shifted left relative to the center of pages.

Author Response

It would be useful to supplement the article with data on surface roughness after AFM (Fig.4).Figures 1 and 3 are shifted left relative to the center of pages.

Response: Thanks for the reviewer’s comment. The surface roughness and grain size had been calculated. The images had been adjusted.

Round 2

Reviewer 2 Report

Thank you for conducting a thorough revision. After reading the text, I believe the article should be released for publication. First, I would still like the authors to highlight a few issues I noticed. I want to draw the authors' attention to a few discussions that may still be worth including in the revisions. Some of them, in my opinion, are not mandatory. Here is a list of them:

1.      "The peak locations, symmetry assignments along with spectral characteristics are alike other studies, and these proved high crystallinity of thin films [43-45]"

I encourage the authors to try fitting the measured spectra in their future work to extract accurate peak parameters and compare and discuss them with the literature.

2.      R phase, a rutile phase

I have a big problem with using the term "rutile phase" in the context of V2O. The term rutile phase means precisely one of the phases of TiO2. Many publications make this mistake, and I understand that the authors have taken it as a leading one. I think terms such as rutile structure or rutile-like structure should instead be used here. This means that the tetragonal variety of V2O has a rutile structure, not a rutile phase.

3.      "The (011) peaks are far powerful compared to others, showing the thin films have a (011) preferred orientation"

I would be careful with such a clear indication of preferential growth direction. The physics of the diffraction phenomenon on crystal planes has it that reflections at low angles always have a high intensity compared to reflections at high angles—moreover, many reflections in the XRD spectrum for layers are obtained by magnetron sputtering. I still think there should be no unscripted reflections left in the spectrum. This could be perceived as something the authors want to keep silent.

Author Response

  1. "The peak locations, symmetry assignments along with spectral characteristics are alike other studies, and these proved high crystallinity of thin films [43-45]"

I encourage the authors to try fitting the measured spectra in their future work to extract accurate peak parameters and compare and discuss them with the literature.

Response: Thanks for the reviewer’s kindly suggestion, we will try fitting the measured spectra in the future work.

  1. R phase, a rutile phase

I have a big problem with using the term "rutile phase" in the context of V2O. The term rutile phase means precisely one of the phases of TiO2. Many publications make this mistake, and I understand that the authors have taken it as a leading one. I think terms such as rutile structure or rutile-like structure should instead be used here. This means that the tetragonal variety of V2O has a rutile structure, not a rutile phase.

Response: Thanks for the reviewer’s comment. We had changed the description in the article. The “rutile phase” and “monoclinic phase” had been replaced by “rutile-like structure” and “monoclinic-like structure”, respectively.

  1. "The (011) peaks are far powerful compared to others, showing the thin films have a (011) preferred orientation"

I would be careful with such a clear indication of preferential growth direction. The physics of the diffraction phenomenon on crystal planes has it that reflections at low angles always have a high intensity compared to reflections at high angles—moreover, many reflections in the XRD spectrum for layers are obtained by magnetron sputtering. I still think there should be no unscripted reflections left in the spectrum. This could be perceived as something the authors want to keep silent.

Response: Thanks for the reviewer’s comment. In order to represent the situation in a better way, the description of “clear indication of preferential growth direction” had been deleted.

“The (011) peaks are far powerful compared to others, this consequence corresponds to others' reports [43,46].”